# Endometrial Cancer with and without Endometriosis: Clinicopathological Differences

**DOI:** 10.3390/cancers15235635

**Published:** 2023-11-29

**Authors:** Takahiro Minamikawa, Nozomi Yachida, Kotaro Takahashi, Kyota Saito, Tomoyuki Sekizuka, Hidehiko Akashi, Miho Suzuki, Yutaro Mori, Kaoru Yamawaki, Kazuaki Suda, Ryo Tamura, Sosuke Adachi, Kosuke Yoshihara

**Affiliations:** 1Department of Obstetrics and Gynecology, Uonuma Kikan Hospital, Niigata 949-7302, Japan; takahiro19780808@gmail.com; 2Department of Obstetrics and Gynecology, Niigata University Graduate School of Medicine and Dental Sciences, Niigata 951-8126, Japan; nyachida@med.niigata-u.ac.jp (N.Y.); k-saito@med.niigata-u.ac.jp (K.S.); tsekiduka@med.niigata-u.ac.jp (T.S.); hakashi@med.niigata-u.ac.jp (H.A.); miho-suzuki1289@med.niigata-u.ac.jp (M.S.); yutmori@med.niigata-u.ac.jp (Y.M.); kymwk@med.niigata-u.ac.jp (K.Y.); sudakazuaki@med.niigata-u.ac.jp (K.S.); ryo.h19@gmail.com (R.T.)

**Keywords:** endometrial cancer, endometriosis, clinicopathological difference

## Abstract

**Simple Summary:**

Previous studies have shown that women with endometriosis have an increased risk of ovarian cancer. However, it is unclear whether endometriosis is associated with a risk of developing endometrial cancer. Therefore, this study was designed to retrospectively assess the clinicopathological relationship between endometrial cancer and endometriosis using the medical records of patients with endometrial cancer who underwent surgery at our institution.

**Abstract:**

Endometriosis is known to be associated with an increased risk of endometrioid and clear cell ovarian cancer. However, the association between endometriosis and endometrial cancer is controversial. Therefore, we retrospectively analyzed the medical records of women with endometrial cancer who had undergone surgery at our institution to evaluate the clinicopathological relationship between endometrial cancer and endometriosis. The study included 720 women pathologically diagnosed with endometrial cancer at our hospital between 2000 and 2020. The participants were allocated to two groups of patients with endometrial cancer: patients with endometriosis (*n* = 101) and patients without endometriosis (*n* = 619). Endometrial cancer patients with endometriosis were significantly younger (median age 54.0 vs. 58.0; *p* = 0.002). In addition, endometrial cancer patients with endometriosis had fewer pregnancies and deliveries (median pregnancy 1.58 vs. 1.99; *p* = 0.019, median delivery 1.25 vs. 1.56; *p* = 0.012). The percentage of patients classified as stage IA was significantly higher in those with endometrial cancer with endometriosis (68.3% vs. 56.4%; *p* = 0.029). In the analysis of synchronous ovarian cancer, the percentage of dual primary cancer was higher in patients with endometriosis (14.9% vs. 1.6%; *p* < 0.001). The association of young-onset early-stage endometrial cancer with endometriosis is an important finding that cannot be ignored clinically.

## 1. Introduction

Endometriosis is defined as the presence and growth of ectopic endometrial tissue outside of the uterine cavity. It is relatively common, affecting an estimated 10–15% of women of reproductive age [1,2,3,4,5]. The cause of endometriosis has not been clearly defined. Nevertheless, several theories have been proposed, including retrograde menstruation, dissemination of endometrial cells through the blood or lymphatic system, and coelomic dysplasia [6]. On the other hand, endometriosis also shares characteristics with malignant tissue, such as tissue invasion, induction of angiogenesis, increased proliferative capacity, ability to evade apoptosis, and ability to form local and distant foci [7,8]. Several studies have established that a history of endometriosis is associated with an increased risk of both endometrioid and clear cell ovarian cancer [9]. However, its association with endometrial cancer remains unclear. Several large population-based studies have shown that women with endometriosis have an increased risk of endometrial cancer [10,11,12], whereas other studies have shown no association [13,14,15,16,17].

Epidemiological risk factors for endometrial cancer include age, exogenous estrogen use, high estrogen status due to obesity, early age at menarche, nulliparity, and late menopause [18]. Endometriosis also has similar risk factors related to estrogen dependency, such as early menarche and nulliparity. Moreover, the origin of both endometriosis and endometrial cancer is the uterine endometrium [19]. Recently, our genomic analysis of endometriosis and normal endometrium revealed the presence of somatic mutations in cancer-associated genes such as *KRAS* and *PIK3CA* in endometriotic and normal endometrial epithelial cells [20,21,22,23,24,25,26]. These cancer-associated genetic mutations are also frequently found in endometrial cancer. Based on the above description, we hypothesized that endometriosis and endometrial cancer share a common etiologic mechanism and that patients with endometriosis have persistent cancer-associated gene mutations in the normal endometrium, making them more susceptible to endometrial cancer. In this study, we aimed to characterize the clinicopathological and molecular biology of endometrial carcinoma with or without endometriosis to determine the significance of endometriosis in the pathogenesis of endometrial carcinoma.

## 2. Materials and Methods

This retrospective cohort study was approved by the institutional review board (2023-0201). This study was conducted in accordance with the Declaration of Helsinki and guidelines for good epidemiological practice.

Clinicopathological data were collected and analyzed from the institutional medical records. A total of 720 patients with endometrial cancer who underwent primary surgery at the Niigata University Medical and Dental Hospital, Japan, between January 2000 and December 2020 were included in this study. Based on stage and histology, the recurrence risk of endometrial cancer was classified as low, intermediate, or high risk, and patients with intermediate risk or higher were basically treated with taxane and platinum-based combination chemotherapy after surgery.

The stage of endometrial cancer was determined by the 2008 FIGO staging system [27]. Age at menopause was defined as age after 12 consecutive months of amenorrhea. Histology of endometrial cancer was diagnosed according to the WHO criteria [28]. We classified endometrioid G1/2 carcinoma as low-grade, endometrioid G3, clear, serous, and carcinoid as high-grade, and the remaining special types as other. Lymph node metastasis was determined based on histopathological examination of the dissected lymph nodes. For cases without lymph node dissection, lymph node metastasis was defined as the presence or absence of enlarged lymph nodes on preoperative imaging (CT and MRI). The criteria for distinguishing metastatic cancer from dual primary cancer were as follows: metastatic carcinoma was diagnosed based on a multinodular ovarian pattern as a major criterion with two or more of the following as minor criteria: small (<5 cm) ovaries, bilateral ovarian involvement, deep myometrial invasion, vascular invasion, and tubal lumen involvement [29]. The features indicative of dual primary cancer included clear histological distinction of tumors of the endometrium and ovary, no or minimal myometrial invasion, absence of lymphovascular space invasion, unilateral ovarian tumor, and/or presence of ovarian endometriosis [30]. 

We used hematoxylin-eosin-stained slides of surgically removed organs to search for the presence of endometriosis pathologically. Only endometrial cancer patients with pathologically diagnosed comorbid endometriotic lesions were considered to have endometriosis.

All standard statistical tests were conducted using R (http://www.r-project.org, accessed on 15 September 2023). Categorical variables were compared between the two groups using Fisher’s exact test, and continuous variables between the two groups were compared using the Wilcoxon rank-sum test. The Kaplan–Meier method was used to estimate the survival distribution. The log-rank test was used to calculate the statistical significance between groups regarding disease recurrence and death. All statistical tests were 2-sided, and statistical significance was defined as a *p*-value of less than 0.05.

## 3. Results

A total of 720 patients with EC were included in this retrospective cohort. All cases were classified into two groups: endometrial cancer with endometriosis (101 patients) and endometrial cancer without endometriosis (619 patients). Details of endometriosis sites coexisting with endometrial cancer are shown in Table 1. Ovarian endometriosis was frequent in endometrial cancer patients with endometriosis.

We compared the clinical characteristics between endometrial cancers with or without endometriosis (Table 2). Intriguingly, endometrial cancer patients with endometriosis were significantly younger than those without endometriosis (median age 54.0 vs. 58.0; *p* = 0.002). In addition, endometrial cancer patients with endometriosis had fewer pregnancies and deliveries (median pregnancy 1.58 vs. 1.99; *p* = 0.019, median delivery 1.25 vs. 1.56; *p* = 0.012). 

Next, we compared the pathological characteristics between endometrial cancer patients with and without endometriosis (Table 3). The percentage of stage IA was significantly higher in endometrial cancer patients with endometriosis (68.3% vs. 56.4%; *p* = 0.029). While the details of each histologic type are shown in Table 3, the frequency of endometrioid histologic type was similar between the two groups (83.1% vs. 83.0%; *p* = 1.0). When endometrioid G1 and G2 were classified as low-grade, endometrioid G3, clear, and serous as high-grade, and the remaining special types as other, no statistical significance in the distribution of histologic types was also observed between the two groups.

In the analysis of synchronous ovarian cancer, there was no significant difference in the percentage of cases with metastasis from endometrial cancer to the ovary between the two groups (Table 4). However, the percentage of dual primary cancers was higher in endometrial cancer patients with endometriosis compared to those without endometriosis (14.9% vs. 1.6%; *p* < 0.001). In 25 dual primary cancer cases, the combinations of histologic types of endometrial and ovarian cancers are shown in Table 5. The pattern of endometrioid histologic type in both endometrial and ovarian cancers was most frequent in the dual primary cancers regardless of endometriosis but the distribution of endometrioid and clear cell histologic types was slightly higher in the patients with endometriosis than those without endometriosis (80% vs. 60%).

In the prognostic analysis, endometrial cancer patients with endometriosis had marginally longer progression-free survival (PFS) compared to those without endometriosis (Appendix A). However, no significant difference in overall survival (OS) was observed between the two groups. Because the distribution of stage IA was higher in the patients with endometriosis, we focused on stage IA. There were no significant differences in PFS and OS between stage IA endometrial cancer patients with and without endometriosis (Appendix A). In addition to stage, ovarian cancer affects prognosis in synchronous cancers if ovarian cancer is advanced. Therefore, we excluded dual primary cancer cases from the analysis dataset and re-performed the prognostic analysis between the two groups. Although endometrial cancer patients with endometriosis tended to have longer PFS but not OS (Figure 1), there was no significant difference in PFS and OS between stage IA endometrial cancer patients with and without endometriosis (Figure 2). 

## 4. Discussion

In many studies, scholars have attempted to find an association between endometriosis and endometrial cancer, but the results remain controversial [10,11,12,13,14,15,16,17]. This study identified the clinicopathological characteristics of endometrial cancer with endometriosis. Our data also suggest the importance of endometriosis in the pathogenesis of endometrial cancer.

Based on clinicopathological characteristics, endometrial carcinomas are classified as type I, which affects approximately 80% of patients, and type II, which affects approximately 20% of patients [31,32]. Type II tumors are predominantly non-endometrioid carcinomas, such as serous or clear cell histological types [33]. On the other hand, Type I tumors arise from atypical glandular hyperplasia. This type is associated with long-lasting, noncompetitive estrogenic stimulation and is often preceded by endometrial hyperplasia [34]. Endometriosis is also an estrogen-dependent disease, as is type I endometrial cancer [35,36]. Endometriosis shares many of the key features, such as resistance to apoptosis, stimulation of angiogenesis, invasion, and inflammation, with cancer [26]. In addition to the same etiologic origin, common clinicopathologic features, including estrogen dependence, may be important in explaining the association between uterine carcinoma and endometriosis. 

In fact, genomic analysis demonstrated several commonalities between endometriosis and type I endometrial cancer. Type I endometrial cancer has few copy number alterations or *TP53* mutations but frequent somatic mutations in *PTEN*, *CTNNB1*, *PIK3CA*, *ARID1A*, *KRAS*, *FBWX7*, and *PPP2R1A* [37]. Our previous study clarified that *PIK3CA*, *ARID1A*, *KRAS*, *FBWX7*, and *PPP2R1A* are also frequently mutated in ovarian endometriosis [23]. Of the common cancer-associated genes, several genes such as *PIK3CA*, *KRAS*, *FBWX7*, and *PPP2R1A* are frequently mutated in uterine normal endometrium, which is the origin of both endometriosis and endometrial cancer. Mutant allele frequencies of these cancer-associated genes are higher in endometriosis or endometrial cancer compared to those in normal endometrium, suggesting clonal expansion of epithelial cells with cancer-associated gene mutations from normal endometrium to endometriosis and endometrial cancer [23,37,38]. In other words, the accumulation of genomic alterations in normal endometrial cells is an important key to associating endometriosis with endometrial cancer. 

In the normal uterine endometrial epithelium, somatic mutations increase with aging or the number of menstruation events but are insufficient for malignant transformation [38,39]. Moore et al. demonstrated that the mutational burden was significantly higher in endometrial cancer than in normal endometrium [38]. The accumulation of further cancer-associated gene mutations combined with microenvironmental factors, such as chronic estrogen exposure and inflammation, can lead to cancer development. In our study, patients with endometrial cancer with endometriosis were significantly younger than those with endometrial cancer without endometriosis (median age 54.0 vs. 58.0; *p* = 0.002). The presence of cancer-associated gene mutations in the normal endometrium of patients with endometriosis may have triggered this fact indicated above. In other words, it may indirectly indicate that the patient is at risk of carcinogenesis in the endometrium at the time of endometriosis. The higher ratio of stage IA endometrial cancer with endometriosis cases may be because patients with endometriosis have more opportunities to consult a doctor or have more experience in seeing a gynecologist, which may have reduced resistance to consulting a doctor and led to early detection of the disease. Another possibility is that the common cancer-related gene mutations found in endometriosis and endometrial epithelium are *PIK3CA* and *KRAS*, which are presumed to reflect a multistep carcinogenesis model in many cases [24]. 

We found that endometrial cancer with endometriosis was associated with fewer pregnancies and deliveries (median pregnancy 1.58 vs. 1.99; *p* = 0.019, median delivery 1.25 vs. 1.56; *p* = 0.012). Endometriosis can lead to infertility [40,41]. The mechanism of infertility in cases of severe invasive endometriosis involves a change in the normal anatomy of reproductive organs [41,42]. However, it is not clear how superficial ectopic endometriotic lesions affect infertility. On the other hand, regarding the relationship between parity and endometrial cancer, there is strong evidence that the incidence of endometrial cancer is reduced by 40% in parous women compared to nulliparous women. Hormonal changes during pregnancy, with increased progesterone production, which has a protective effect on the endometrium, may explain this association [31]. In other words, the risk of endometrial cancer may have increased due to the reduced number of pregnancies and reduced ability to protect the endometrium as a result of endometriosis. 

Endometriosis has been found to be associated with several histologic subtypes of epithelial ovarian cancer known as endometriosis-associated ovarian cancer, such as clear cell and endometrioid carcinomas, which are etiologically distinct in several aspects from other subtypes of ovarian cancer [43]. Numerous studies have shown that endometriosis is associated with the risk of developing clear cell and endometrioid carcinomas [44]. The frequency of malignant transformation of ovarian endometriosis is estimated to be 0.7% [45] and 4.2 times the risk of developing ovarian cancer [46], and a study by Zaino et al. confirmed that in synchronous endometrial and ovarian cancers, approximately 30% of patients had endometriosis [47]. The results of our study support this finding. In fact, the percentage of dual primary cancers was higher in patients with endometriosis (14.9% vs. 1.8%; *p* < 0.001) in this study. In dual primary cancer patients with endometriosis, all endometrial cancers were endometrioid histologic type and 80% of ovarian cancer was diagnosed as endometrioid or clear cell histologic types, so-called endometriosis-associated ovarian cancer. This may reflect that the presence or absence of endometriosis is used as a reference in diagnosing metastasis or dual primary cancer in endometrial and ovarian cancer.

In addition to endometriosis, which is the presence of endometrial-like glands and stroma outside the uterine cavity, benign gynecological diseases include adenomyosis, which is the presence of endometrial-like glands and stroma within the myometrium [48]. Hermens et al. published a study including around 13,000 women, which showed that women with endometriosis or adenomyosis have an increased incidence of endometrial cancer which may indicate a possible association between endometriosis/adenomyosis and endometrial cancer [49]. Therefore, caution may be required for both endometriosis and adenomyosis.

There are some limitations to this study. The first issue is how to confirm whether endometriosis coexists with endometrial cancer. In this study, we preoperatively suspected endometriosis lesions through transvaginal ultrasonography, MRI, and CT, and intraoperatively looked for endometriosis lesions. Then, we histopathologically confirmed the presence or absence of endometriosis lesions using the surgical specimen. However, very small endometriosis lesions in the peritoneum may be missed. Indeed, the frequency of peritoneal endometriosis is lower than that of ovarian endometriosis (Table 1). It is difficult to prove that endometriosis does not exist in patients with endometrial cancer.

The next issue is that in the analysis of synchronous cancers, dual primary cancers or metastasis to the ovary were diagnosed by histopathologic criteria [29,30]. In the synchronous endometrial and ovarian cancers, methods to define independent primary tumors or metastatic tumor remains controversial. Some genomic studies suggested that many synchronous endometrial and ovarian cancers originated from a single tumor [50,51,52]. However, these studies did not consider genomic alterations of normal uterine endometrium as the origin of both endometrial cancer and endometriosis. A recent study demonstrated the differences in DNA methylation signatures between endometrial and ovarian cancers [53]. Therefore, genomic/epigenomic analyses from endometrial and ovarian cancer sites, as well as normal endometrium and endometriosis, may be helpful to accurately diagnose whether dual primary cancers were present or not. The last issue is that the evidence derived from a retrospective cohort study is generally lower in research quality than that from a prospective study. To this end, further prospective studies with adequate sample sizes are needed to verify the temporal association between endometriosis and endometrial cancer.

## 5. Conclusions

Our study suggested that regular surveillance for the early detection and diagnosis of endometrial cancer might be needed for women with endometriosis. Further studies are needed to clarify the relationship between endometriosis and endometrial cancer.

## Figures and Tables

**Figure 1 cancers-15-05635-f001:**
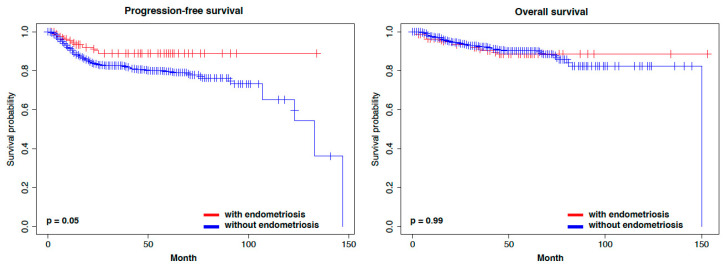
Kaplan–Meier curves representing the survival in relation to all stages with and without endometriosis.

**Figure 2 cancers-15-05635-f002:**
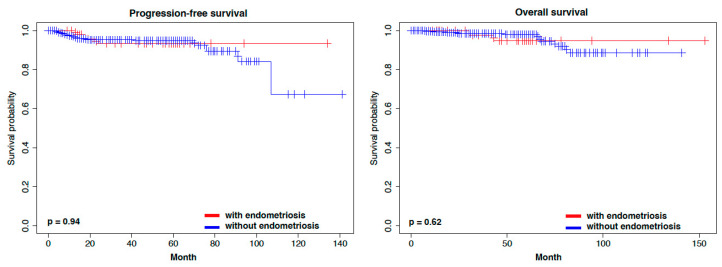
Kaplan–Meier curves representing the survival in relation to stage IA with and without endometriosis.

**Table 1 cancers-15-05635-t001:** Details of endometriosis sites coexisting with endometrial cancer.

Site of Endometriosis	Case *
Ovary	48.5% (49/101)
Uterus	48.5% (49/101)
Fallopian tube	4.0% (4/101)
Peritoneum	2.0% (2/101)
Lymph node	2.0% (2/101)
Colorectal	1.0% (1/101)

* Cases are duplicated.

**Table 2 cancers-15-05635-t002:** Comparison of clinical characteristics between endometrial cancer with and without endometriosis.

	With Endometriosis (101)	Without Endometriosis (619)	*p*-Value
Age	54.0 (31.0–76.0)	58.0 (22.0–94.0)	0.002
BMI	24.6 ± 5.6	24.6 ± 5.6	0.852
CA125 [IU]	219.1 ± 778.8	98.9 ± 344.6	0.172
Pregnancy	1.58 ± 1.36	1.99 ± 1.54	0.019
Delivery	1.25 ± 1.09	1.56 ± 1.15	0.012
Hypertension	23.8% (24/101)	26.7% (166/618)	0.722
Diabetes Mellitus	15.8% (16/101)	12.9% (80/618)	0.534
Menopause	56.4% (57/101)	63.3% (392/619)	0.539

**Table 3 cancers-15-05635-t003:** Comparison of pathological characteristics between endometrial cancer with and without endometriosis.

	With Endometriosis	Without Endometriosis	*p*-Value
Number of cases	101	619	-
Stage			
IA	68.3% (69/101)	56.4% (349/619)	0.029
IB	8.0% (8/101)	14.9% (92/619)	
II	6.0% (6/101)	7.8% (48/619)	
IIIA	5.0% (5/101)	4.0% (25/619)	
IIIB	2.0% (2/101)	0.5% (3/619)	
IIIC1	3.0% (3/101)	5.7% (35/619)	
IIIC2	1.0% (1/101)	4.0% (25/619)	
IVA	0 (0/101)	0.2% (1/619)	
IVB	7.0% (7/101)	7.0% (41/619)	
Histology			
Endometrioid	83.1% (84/101)	83.0% (514/619)	1.0
grade 1	58.4% (59/101)	52.8% (327/619)	
grade 2	18.8% (19/101)	22.0% (136/619)	
grade 3	5.9% (6/101)	8.2% (51/619)	
Clear	5.9% (6/101)	3.2% (20/619)	
Serous	4.0% (4/101)	4.7% (29/619)	
Mucinous	1.0% (1/101)	0.2% (1/619)	
Mixed	4.0% (4/101)	4.0% (25/619)	
Carcinosarcoma	1.0% (1/101)	3.1% (19/619)	
Others	1.0% (1/101)	1.8% (11/619)	

**Table 4 cancers-15-05635-t004:** Details of synchronous ovarian and endometrial cancers with and without endometriosis.

Synchronous Ovarian Cancer	With Endometriosis (101)	Without Endometriosis (619)	*p*-Value
Metastasis	6.0% (6/101)	6.1% (38/619)	1.0
Dual primary	14.9% (15/101)	1.6% (10/619)	<0.001

**Table 5 cancers-15-05635-t005:** Summary of dual primary cancers (*n* = 25) with and without endometriosis.

Endometrial Cancer	Ovarian Cancer	With Endometriosis (*n* = 15)	Without Endometriosis (*n* = 10)
Endometrioid	Endometrioid	73.3% (11/15)	50.0% (5/10)
Endometrioid	High-grade serous	13.3% (2/15)	10.0% (1/10)
Endometrioid	Clear	6.6% (1/15)	10.0% (1/10)
Endometrioid	Mixed	6.6% (1/15)	10.0% (1/10)
Mixed	High-grade serous	0.0% (0/15)	10.0% (1/10)
Mixed	Mucinous	0.0% (0/15)	10.0% (1/10)

## Data Availability

The data that support the findings of this study are available from the corresponding authors, S.A., K.Y. (Kosuke Yoshihara) upon reasonable request.

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
