# Peer review of "Endometrial Cancer with and without Endometriosis: Clinicopathological Differences"

_cancers, 2023, doi:10.3390/cancers15235635_

Round 1
Reviewer 1 Report
Comments and Suggestions for Authors
In this well-written manuscript, the authors explore the association of endometriosis with endometrial cancer and clinicopathologic findings of these cases. The authors conduct a well-described retrospective cohort study that ultimately demonstrates an increased rate of synchronous ovarian cancer in cases of endometriosis and endometrial cancer, as compared to cases of endometrial cancer without endometriosis. My only comment would be to adjust the verbiage behind "double cancer" as it is used throughout the paper. It makes it hard to ascertain if the authors are talking about metastatic disease (IIIA), dual primaries/synchronous disease, or both. Other minor comments are highlighted below.
Line 56 -- The author states that the origin of endometriosis is endometrium. Please provide citations for this, as whether the endometriotic tissue arises from the endometrium or de-novo is part of the question. The tissues do possess similar histologic findings.
Line 119 -- CA125 is also associated with metastatic endometrial cancer, and should include a citation.
Line 240 -- Authors should recognize the retrospective nature of this study as a limitation.
Author Response
Comment
In this well-written manuscript, the authors explore the association of endometriosis with endometrial cancer and clinicopathologic findings of these cases. The authors conduct a well-described retrospective cohort study that ultimately demonstrates an increased rate of synchronous ovarian cancer in cases of endometriosis and endometrial cancer, as compared to cases of endometrial cancer without endometriosis. My only comment would be to adjust the verbiage behind "double cancer" as it is used throughout the paper. It makes it hard to ascertain if the authors are talking about metastatic disease (IIIA), dual primaries/synchronous disease, or both. Other minor comments are highlighted below.
Response:
We apologize for our confusing expression. We have changed the phrase "double cancer" used in the paper to "dual primary cancer" as the reviewer mentioned.
Comment
Line 56 -- The author states that the origin of endometriosis is endometrium. Please provide citations for this, as whether the endometriotic tissue arises from the endometrium or de-novo is part of the question. The tissues do possess similar histologic findings.
Response:
We appreciate the reviewer`s comment. We have added which reference we have cited from (line 59).
Comment
Line 119 -- CA125 is also associated with metastatic endometrial cancer, and should include a citation.
Response:
We appreciate the reviewer`s comment. In our data, we were not able to show that CA125 was significantly higher in endometrial cancer with endometriosis. To avoid confusing expression, we removed it from our paper (line 120).
Comment
Line 240 -- Authors should recognize the retrospective nature of this study as a limitation.
Response:
We appreciate the reviewer`s comment. We have added the descriptions that evidence derived from a retrospective cohort study has a limitation in the revised manuscript (lines 277-281).
Reviewer 2 Report
Comments and Suggestions for Authors
The authors describe; Endometrial cancer with and without endometriosis: Clinico-pathological differences.
The manuscript is well written and easy to follow. The group of patients studies is sufficiently large and the results are interesting.
What I miss in the mamuscript is a more detailed description of the endometriosis, especially the internal endometriosis otherwise called adenomyosis. Some studies explicitly have shown that adenomyosis is associated with endometrial cancer while endometriosis has an association which is much weaker. (ref: Hermens et al. Cancers (Basel). 2021 Sep 13;13(18):4592.) I do not find any reports on adenomyosis in this manuscript which makes it less valuable for clinical use. The authors suggest surveillance of patients with endometriosis, however they may confer this to women with endometriosis and/or adenomyosis.
Author Response
Comments
The authors describe; Endometrial cancer with and without endometriosis: Clinico-pathological differences.
The manuscript is well written and easy to follow. The group of patients studies is sufficiently large and the results are interesting.
Response:
We deeply appreciate the reviewer`s positive evaluation on our manuscript and provided valuable comments to improve the quality of our manuscript.
Comments
What I miss in the manuscript is a more detailed description of the endometriosis, especially the internal endometriosis otherwise called adenomyosis. Some studies explicitly have shown that adenomyosis is associated with endometrial cancer while endometriosis has an association which is much weaker. (ref: Hermens et al. Cancers (Basel). 2021 Sep 13;13(18):4592.) I do not find any reports on adenomyosis in this manuscript which makes it less valuable for clinical use. The authors suggest surveillance of patients with endometriosis, however they may confer this to women with endometriosis and/or adenomyosis.
Response:
We appreciate the reviewer`s comment. According to the reviewer’s comments, we have added the publication that the reviewer introduced, and discussed the association between adenomyosis and endometrial cancer in the revised manuscript (lines 250-257).
Round 2
Reviewer 2 Report
Comments and Suggestions for Authors
the manuscript has been revised well.